# Over 20-Year Follow-up of Patients with Hepatic Glycogen Storage Diseases: Single-Center Experience

**DOI:** 10.3390/diagnostics10050297

**Published:** 2020-05-13

**Authors:** Edyta Szymańska, Patryk Lipiński, Dariusz Rokicki, Janusz Książyk, Anna Tylki-Szymańska

**Affiliations:** 1Department of Gastroenterology, Hepatology, Feeding Disorders and Pediatrics, Children’s Memorial Health Institute, 04-730 Warsaw, Poland; edyta.szymanska@onet.com.pl; 2Department of Pediatrics, Nutrition and Metabolic Diseases, Children’s Memorial Health Institute, 04-730 Warsaw, Poland; p.lipinski@ipczd.pl (P.L.); d.rokicki@ipczd.pl (D.R.); j.ksiazyk@ipczd.pl (J.K.)

**Keywords:** glycogen storage disease, long-term follow-up, disease outcome, hypertension, hepatocellular adenoma, short stature

## Abstract

Background: The published data on the long-term outcomes of glycogen storage disease (GSD) patients is sparse in the literature. The aim of this study was to analyze the long-term (over 20 years) follow-up of patients with hepatic types of GSD-I, III, VI, and IX—from childhood to adulthood, managed by one referral center. Patients and methods: Thirty adult patients with hepatic GSD were included in the study. A retrospective chart review of patients’ medical records has been performed. Results: During the long-term follow-up, the most frequent complications observed in a group of 14 GSD I patients were nephropathy with blood hypertension (10/14), hyperuricemia (8/14), and development of hepatocellular adenomas (HCA; 5/14). All individuals but four presented with normal height. Two patients with GSD Ib suffered from inflammatory bowel disease (IBD). Nine (64%) GSD I patients were in balanced metabolic condition at the age of 18. Regarding GSD III/VI/IX, the most frequent complication was short stature observed in 5 out of 16 patients. All patients but one with GSD VI were in balanced metabolic condition at the age of 18. Conclusion: The long-term outcomes of patients with GSD depend mainly on proper (adjusted to each type of GSD) dietary management and patient compliance. However, in GSD type I, even proper management does not eliminate all long-term complications in adulthood.

## 1. Introduction

Glycogen storage disorders (GSD) are rare inborn errors of carbohydrate metabolism; there are eight liver types: Ia and b, III, IV, VI, IX, XI, and 0, respectively. Each type is a distinct entity with a distinct metabolic block (deficiency of a different enzyme). 

### 1.1. GSD Type I

GSD type I is caused by the deficiency of glucose-6-phosphase (G6P in subtype Ia) or G6P transporter/translocase through the mitochondrial membrane (G6PT in subtype Ib) [1]. That is a key enzyme in the gluconeogenesis and glycogenolysis processes, and therefore GSD I is the most severe type. Decreased/lack of G6P activity leads to accumulation of glycogen and fats within hepatocytes, kidneys and erythrocytes. The early-onset symptoms include hepatomegaly and hypoglycemia with increased lactate (LA) levels, which occurs soon after the meal (within 3–4 h). Patients may be of short stature. Typical biochemical abnormalities include hyperlipidermia and hyperuricemia [2]. Patients with subtype Ib present with neutropenia, which causes recurrent bacterial infections [3], and thus they are treated with granulocytes colony stimulating factor (G-CSF). Since G6P is an enzyme present not only in the liver, but also in the kidneys, blood hypertension and nephropathy may occur. Vitamin D deficiency and osteopenia/osteoporosis are also common complications of GSD I [4,5]. 

Since maintaining a stable blood glucose concentration is a goal of the therapy, uncooked cornstarch (CS) is a mainstay of nutritional treatment in type I, in which hypoglycemia occurs faster than in types III/VI/IX, and these episodes are more severe because ketogenesis has not the time to be triggered, the fasting tolerance is being too short [2]. However, it seems that management with diet does not eliminate late-onset renal damage, nor subsequent renal failure, nephrolithiasis, and development of hepatocellular adenomas (HCA); and an increased risk for HCC (Hepatocellular carcinoma) still remains as a long-term complication, especially in GSD type I [6,7]. 

### 1.2. GSD Type III, VI, and IX

These are so-called ketotic types since ketone production during hypoglycemia is higher than in type I. GSD type III is caused by decreased/lack of activity of debranching enzyme; type VI is caused by glycogen phosphorylase inactivity; and in type IX phosphorylase kinase (pK) is dysfunctional. These GSDs are characterized by hepatomegaly and episodes of hypoglycemia, which are less severe than in type I and their frequency decreases with age [8,9]. 

In type III, cardiomegaly and cardiomyopathy may occur, and creatine kinase (CK) level is usually increased. Type VI is clinically the mildest one, and some adults are almost asymptomatic. GSD IX includes four subtypes and the most common one is an X-linked form (GSD IX-alpha), accounting for 75% of all cases. 

The symptoms include hepatomegaly, mild psychomotor retardation, short stature, and mild muscle hypotonia. The patients’ symptoms become milder with age [10]. In GSD III/VI/IX, a high-protein diet is provided; GSD type III patients often need cornstach when they are children [11]. Since type IIIa and IX-alpha present also with muscular manifestations (myopathy, cardiomyopathy), the high protein diet is the basis of dietary management [12]. 

### 1.3. Diagnosis

The diagnosis of GSD is established based on the patient’s clinical phenotype and the results of biochemical analyses, and is finally confirmed (if available) by molecular analyses.

Typical symptoms common for all GSD types but one (type 0) are hepatomegaly (not present in type 0), hypoglycemia, short stature, and frequently, due to the dietary management with CS—obesity (typically in type I) [3,8,9].

Biochemical abnormalities include hypoglycemia, hyperlipidermia, hyperuricemia (in type I), elevated serum transaminases, increased LA level (type I), increased CK level (type III, IX), and neutropenia in type Ib [2,10].

Prolonged fasting tests are not usually performed and are not useful (and even dangerous) for differentiating between type I and type III. Enzymatic assays are no longer performed, and they have been replaced by molecular analyses. 

Furthermore, abdominal ultrasonography and transient elastography by FibroScan (if available) are performed to assess hepatomegaly and nephromegaly; the liver’s echostructure and its steatosis/fibrosis state; and the presence of any liver lesions. 

Various diagnostic tools are used to detect GSD’s complications. In our center, anthropometric assessment and calorimetry are available and are performed to analyze each patients’ nutritional status. Densytometry is provided to assess each patient’s skeletal condition. In order to detect HCA/HCC, both CT and MRI scans are used, and AFP (Alpha-fetoprotein) level is regularly checked.

### 1.4. Medical Management

The mainstay of medical management in GSD is to prevent episodes of hypoglycemia and to maintain stable blood glucose level; therefore, dietary treatment is the basic therapy. Specific dietary interventions for each type are mentioned above. Nonetheless, the disease complications, if present, need some additional management, including allopurinol in patients who develop hyperuricemia, ACE-inhibitors in patients with blood hypertension, 5-ASA in GSD Ib patients who develop inflammatory bowel disease (IBD), G-CSF in GSD Ib patients with neutropenia, and hypolipemic agents in patients with hyperlipidemia.

The aim of this study was to analyze the long-term outcomes (over 20 years) of patients with hepatic types of GSD-I, III, VI, and IX—from childhood to adulthood, managed by one referral center. This is a unique opportunity, since our group is large for a rare disease, and the monitoring and follow-up have been provided by one site only.

## 2. Patients and Methods

Thirty adult patients with hepatic GSDs, including 8 with GSD Ia, 6 with GSD Ib, 8 with GSD III, 7 with GSD VI, and 1 with GSD IX, were included in the study. The observation period was between 1982 and 2018. Among 30 patients diagnosed and treated since infancy, 25 (83%) have still followed-up with and been monitored by our Institute (Children’s Memorial Health Institute—CHMI) within the outpatient clinic since the age of 18.

Based on medical charts of patients, the following parameters were analyzed: signs and symptoms at time of diagnosis; age at diagnosis; dietary and medical management; anthropological and biochemical parameters, including body mass index (BMI) and serum aspartate aminotransferase (AST), serum alanine aminotransferase (ALT), serum triglyceride (TG), serum total cholesterol (TC), lactate (LA), and uric acid (UA) levels; and long-term complications.

Ethical approval (Code 167/KD/2014, date 29/October/2014) of the study protocol was obtained from the Children’s Memorial Health Institute Bioethical Committee, Warsaw, Poland. 

## 3. Results

### 3.1. GSD Type Ia and Ib

There were 14 patients with GSD type I, including eight individuals with type Ia and six with type Ib. The majority (5/8) of patients with type Ia were diagnosed within their first year of life (mean age: 1.5 years old, age range: 5 months old to 4 years old), and the mean age of GSD Ib diagnosis was approximately 3 (age range: 6 months old to 7 years old) (Table 1). 

All patients had dietary management (individually assessed energy requirements and proportions of diet components) introduced once the diagnosis was established, and CS consumption commenced since the age of 6–8 months (to prevent diarrhea when the intestinal microbiota is not fully developed). Only one GSD Ia patient had nightly glucose infusions for his first year of medical management, which were then substituted with CS. 

Over half of the patients (9/14) were in balanced metabolic condition at the age of 18 (Table 1). Serum AST was elevated in 5/6 GSD Ia patients, and ALT was elevated in three of them. All GSD Ia patients present with elevated TC and TG levels. Serum lactate was elevated in all GSD Ia patients, and uric acid in 3/8 patients.

Serum AST and ALT were within normal laboratory values in all GSD Ib patients. Elevated TG levels were present in 3/5 patients, and TC levels were within normal values in all of them. Serum lactate was elevated in 3/4 patients, and uric acid in 1/5 patients.

The majority of GSD I patients have been followed at our Metabolic Outpatient Clinic, where the diet was supervised and biochemical parameters monitored—in most cases both the compliance and laboratory results were satisfactory (Table 1). During the long-term follow-up, the most common complications observed in GSD I patients were nephropathy with blood hypertension (10/14), hyperuricemia (8/14), and development of HCA (5/14). All individuals but four (29%) were of normal height. 

Almost all patients with GSD type I who developed hyperuricemia were given allopurinol to prevent arthritis. We tried to manage hyperlipidemia with dietary interventions; however, if TG level was above 500 mg/dl, which according to literature [13] is associated with significantly increased risk of acute pancreatitis and HCA/HCC development, the patient usually had hypolipemic agents (either fibrates or statins) introduced. Patients with blood hypertension had ACE-inhibitors applied to prevent progression to an end-stage renal disease. The majority of GSD Ib individuals were treated with G-CSF and those who developed IBD had, additionally, 5-ASA agent introduced. 

The majority of those complications developed either during childhood or adolescence, although the proper diet had been introduced since each diagnosis had been established. Two women with GSD Ia (patient 4 and 5) had menorrhagia; one gave birth to a healthy child without any complications. One patient with GSD Ib (patient 10) developed hearing problems (sensorineural hearing loss). One GSD Ia patient (patient 2) presented with bleeding problems (excessive epistaxis). Two patients with GSD Ib (patient 11 and 12) suffered from IBD, and patient 12 had arthritis. Both patients had active IBD at the age of 18 and have experienced a lot of flare-ups during adulthood. 

### 3.2. GSD Type III, VI, IX

There were 16 patients diagnosed with GSD III/VI/IX: 8, 7, and 1, respectively. In comparison to patients with GSD I, the diagnosis was established later—the mean age at diagnosis was approximately 3 (age range: 4 months old to 8 years old).

All patients but one had balanced metabolic conditions at the age of 18 (Table 2). Serum AST was elevated in 4/8 GSD III patients, in 1/4 GSD VI patients, and in 1 GSD IX patient. Serum ALT was elevated in 3/8 GSD III patients; all GSD VI and GSD IX patients had normal ALT levels. Elevated TC levels were present in 7/14 patients:—4/7 with GSD III, 1/4 with GSD VI, and 1 patient with GSD IX. Elevated TG levels were present in 5/14 patients—3/7 with GSD III and 2/4 with GSD VI. Serum lactate was checked only in one patient (patient 11), and it was within normal values. Uric acid was checked in only 5/14 patients; in all of them it was normal. CK activity was found to be elevated in 3/8 GSD III patients.

During the long-term follow-up the most common complication in these patients was short stature, present in 5/16 patients; in three of them it was a considerably short stature (>2 SD; 31%). One patient with type VI (patient 11) developed diabetes mellitus (DM) type 1 at the age of 4 and was treated with insulin. 

## 4. Discussion

The manuscript presents data from a 20-year follow-up of 30 patients with hepatic GSDs. So far, data published on the long-term outcomes of hepatic GSD patients is sparse. According to the natural course of GSD I, complications develop with time (the older the patient, the more complications that may occur), and include short stature, renal disorders (glomerular hyperfiltration, microalbuminuria, proteinuria, proximal and/or distal tubules dysfunction, and nephrolithiasis), and increased risk for HCA and HCC development [13,14,15]. Neutropenia and neutrophils dysfunction predispose to recurrent bacterial infections and development of IBD in GSD Ib. Patients with GSD Ia have the highest risk of HCA development, and that is rarely observed in GSD III/VI/IX [7]. Childhood hepatic symptoms in GSD III/VI/IX tend to become less severe with age, however (up to 12–15% of adults develop liver fibrosis and cirrhosis) [10].

In our cohort, all patients had proper dietary management (CS in type I, and high protein diet in types III/VI/IX) introduced as soon as the diagnosis of GSD was established. In GSD I patients, CS was introduced at the age of at least 6 months old to prevent diarrhea. Patient compliance was generally satisfactory. Moreover, all of them closely and regularly followed-up, even as adults. Due to that, the patients’ adherence improved (they are taught how to manage the diet and control themselves), and by transition (at the age of 18 when pediatric healthcare can no longer be provided, and internal/adult medicine care is delivered) the vast majority of them had balanced metabolic homeostasis. Such nutritional management prevents hypoglycemia, and thus leads to normalization of biochemical markers (levels of TC and TG; liver function tests—AST and ALT; and uric and lactic acid levels) [16].

In our study, nephropathy with hypertension was the most common complication in GSD I patients. The European study for GSD reported in 2002 that all GSD Ia patients developed microalbuminuria or proteinuria by 24 years of age [17]. Once microalbuminuria developed, treatment with ACE-inhibitors was introduced to prevent progression to end-stage renal disease. No patient in our cohort developed the end-stage renal disease.

In the reported cohort, HCA was diagnosed in five patients with GSD Ia and was one of the most common complications in these patients. In all of those patients, they developed it at adolescence, and it was associated with high serum levels of TG (>500 mg/dL), which is consistent with the literature. HCA formation was one of the first complications reported in GSD type I, but also in type III. Initial studies on GSD type I adults have demonstrated that HCA developed in over 70% of those patients [18]. The lesions typically occurred during puberty, and malignant transformation risk has been observed. Wang et al. have demonstrated that patients achieving TG concentrations under 500 mg/dL have a significantly lower rate of HCA formation [13]. Recently, Beegle et al. have reported that regression of HCA identified with both ultrasound and magnetic resonance imaging (MRI) could occur with improvement in metabolic homeostasis [19]. Not only did adenomas regress with good control, but six of nine patients with serial MRI scans had complete disappearance of lesions after a mean of 4.8 years. One of our female patients with GSD Ia with multiple HCA (Patient 6) has confirmed this observation. Once she improved her compliance with the proper diet and CS intake as an adult, her biochemical markers, including TG level, normalized; she achieved balanced metabolic homeostasis and the lesion within her liver disappeared. Malignant transformations of adenomas have never been observed in our group.

One GSD Ia patient presented bleeding problems (excessive epistaxis), which may occur in GSD type I. The impaired coagulation in type I GSD is due to dysfunctional platelets function.

Inflammatory bowel disease (IBD) due to neutropenia and impaired neutrophils function is a common complication in GSD Ib [20]. The treatment consists of recombinant human granulocyte-colony-stimulating factor (G-CSF), 5-aminosalicylates (ASA), probiotics, and also high dose of vitamin E. In our group, two GSD Ib siblings developed IBD [21]. One of them present with severe cases of bowel disease and arthritis. Both of them have been treated with G-CSF and 5-ASA. Due to lack of proper adult gastroenterological care, their IBD is poorly controlled now. The issue of transition is crucial and very difficult. Although we follow-up on our patients from a metabolic point of view, finding other specialists (gastroenterologists, nephrologists) who would like to treat adult metabolic patients is challenging.

The summary of GSD Ia’s outcome in adults by Weinstein et al. has demonstrated that complications in GSD Ia can be delayed or prevented with a properly balanced diet [22]. Our results and observations are consistent with those presented by Weinstein et al.

Short stature was observed both in type I and in ketotic types, but it was not as a common complication as we may expect, which only confirmed the proper management of these patients by our site.

The important, although rather obvious observation, is that there are significant differences in both short and long-term complications and outcomes between GSD type I and types III/VI/IX. In our group, no patient with type IIIa developed cardiomyopathy, and the only significant complication in these patients was short stature. Peripheral myopathy may occur in GSD III. Although few patients in our group had increased CPK levels, none of them developed miopatic disorders. Patients with GSD III with normal CPK levels were almost asymptomatic.

The majority of individuals with ketotic GSD types, especially with type VI, were asymptomatic, and they did not even consider themselves to be ill when becoming adults. However, most patients followed their diets once educated at infancy; they stuck to the management they got used to, unlike patients with type I. Although the proper diet was introduced, some complications could not be prevented. In our cohort, the majority of GSD I complications developed either during childhood or adolescence. In some cases, it was due to unsatisfied compliance, but in the majority of the patients it was due to the natural course of the disease. This observation demonstrates that type I is much more severe than types III/VI/IX, and unlike them, diet, though it improves the outcome, does not eliminate some complications.

The advantages of our work are a significant number of patients, long-term follow-up, and proper, up-to-date clinical management provided by a single-referral center.

## 5. Conclusions

1. Long-term outcomes of patients with GSD depend mainly on proper (adjusted to each type of GSD) dietary management and patient compliance. However, in GSD type I, even proper management does not eliminate all long-term complications in adulthood.

2. Nephropathy with hypertension was the most common complication in our cohort of GSD I patients.

3. HCA was diagnosed in 5 out of 14 patients with GSD Ia, and it was one of the most common complications in these patients, associated with high levels of serum TG (>500 mg/dL).

4. Regarding GSD III/VI/IX, the quite common complication was short stature, present in 5/16 patients. All patients but one with type VI were in balanced metabolic condition at the age of 18.

## Figures and Tables

**Table 1 diagnostics-10-00297-t001:** Detailed characteristics of adult patients with GSD type I included in the study (*n* = 14).

GSD Ia										
Patient’s Number and Gender	Age at Diagnosis/ Current Age (m—Months, y—Years)	Diet	Laboratory Results at 18 y						Complications and Age of Appearance	Metabolic Homeostasis at Present
			BMI (kg/m^2^)	ALT/AST (U/L)	LA (mg/dL)	TG (mg/dL)	TC (mg/dL)	UA (mg/dL)		
1. M	12 m/46 y	Followed	23.2	36/43	76.5	2190	544	7.8	HCA/KS/HT/HL/HU and gout; since adolescence	Moderate (HL and HT)
2. F	6 m/19 y	Followed	23.7	21/21	65	563	325	5.8	IBD/HT/Recurrent nose bleeding; since childhood and adolescence	Moderate (HL and HT)
3. M	12 m/25 y	Followed; Additional glucose intake (drinks)	N/A	N/A	N/A	N/A	N/A	N/A	N/A	N/A
4. F	5 m/24 y	Followed	23.3	42/60	40	919	439	5.8	HT/HL/M; Since adolescence and adulthood	Moderate (HL and HT)
5. F	3 y/43 y	Followed	22.6	67/68	116	1470	342	9.3	M/HCA/HL/HU; Since adolescence and adulthood	Unbalanced
6. F	6 m/35 y	Followed	N/A	N/A	N/A	>1000	N/A	N/A	HCA/HL/HT; Since adolescence	Due to improvement in patient’s compliance the metabolic homeostasis is stable now/ HCA disappeared
7. M	4 y/43 y	Followed	22.6	62/23	81	219	177	8.3	RH/HCA/HU; Since adolescence	Balanced/ Born a healthy child
8. M	2 y/31 y	Followed	22.3	70/42	116	723	307	8.2	SS/HU/HL; Since childhood	Moderate/ Due to non-compliance HCA/HT/HU/ HL developed
**GSD Ib**											
9. M	1.5 y/31 y	Followed	N/A	N/A	N/A	N/A	N/A	N/A	IgA-N/HT/Neu (800/uL), Neupogen introduced since 13 y (AP before that)/ Since childhood	Moderate
10. M	10 m/24 y	Followed	N/A	6/11	N/A	52	88	5.5	Neu (600/uL)/Recurrent mouth and gingival infections/Sensorineural hearing loss (hearing aids)/HT/AP/ since childhood	Balanced
11. F	6 y/27 y	Followed	20	26/26	28	259	109	8	IBD/HT/HU/ Neu (1300/µL), Neupogen introduced since 10 y (AP before that)/ Since childhood	Moderate/ Recurrent IBD flares at adulthood
12. M	6 m/26 y	Followed	17	7/13	139	170	78	5.3	IBD, HT/Arthritis/Neu (600/µL), Neupogen since 8 y. (AP before that)/ Since childhood	Unbalanced/ Recurrent IBD flares at adulthood, cannot walk due to arthritis, depression
13. F	N/A 47 y	Followed	N/A	8/22	30	193	167	6.7	Recurrent infections due to Neu, only AP, Neupogen has never been administered/ Since childhood	Moderate
14. F	7 y/40 y	Followed	39.6	29/27	8	36	65	6.9	KS/ since adolescence	Balanced/ gave birth to a healthy child

Abbreviations: BMI—body mass index; LA—lactate acid; TG—triglycerides; TC—total cholesterol; UA—uric acid; HCA—hepatocellular adenoma; IBD—inflammatory bowel disease; CS—cornstarch; Neu—neutropenia; N/A—not available; KS—kidney stones; HT—hypertension; HL—hyperlipidemia; HU—hyperuricemia; O—overweight; SS—short stature; M—menorrhagia; AP—antibiotic prophylaxis. Reference values: ALT < 45 U/L; AST < 35 U/L; LA 4.5–19.8 mg/dL; TG < 90 mg/dL; TC < 170 mg/dL; UA 3.4–7.0 mg/dL. Legend: GSD I diet: regular meals/diet without simple sugars/CS regularly during the day and at night; high-carbohydrates and low-fat diet: 60–70% calories from carbohydrates, 10–15% calories from protein, and the remaining calories from fats; high-protein diet: 2–3 g of protein/kg; moderate metabolic homeostasis: mild-moderate biochemical abnormalities, no severe organic complications.

**Table 2 diagnostics-10-00297-t002:** Detailed characteristics of adult patients with GSD types III/VI/IX included in the study (*n* = 16).

Patient’s Number and Gender	Age at Diagnosis/ Current Age (m—Months, y—Years))	Diet	Laboratory Results at 18 y						Complications and Other Issues	Metabolic Homeostasis at Present
			BMI (kg/m^2^)	ALT/AST (U/L)	LA (mg/dL)	TG (mg/dL)	TC (mg/dL)	UA (mg/dL)		
**GSD type III**										
1. M	1.5 y/24 y	Followed	18.5	46/105	N/A	74	153	5.4	None; CK normal	Balanced
2. M	4 y/25 y	Followed	18.7	39/22	N/A	81	179	N/A	None; CK normal	Balanced
3. M	3 y/26 y	High-CH at the beginning then GSD III/VI/IX	20.5	41/29	N/A	103	158	N/A	None; CK normal	Balanced
4. F	2 y/26 y	Followed	22.3	21/25	N/A	83	227	N/A	None; CK elevated (392 U/L)	Balanced
5. F	6 y/33 y	Followed	24.4	36/22	N/A	N/A	N/A	N/A	None; CK normal	Balanced
6. F	10 m/36 y	Followed	23.2	79/84	N/A	86	206	6.6	Back pain, fatigue; CK elevated (1336 U/L)	Balanced
7. F	3 y/37 y	Followed	24.6	64/44	N/A	220	188	N/A	Fatigue; CK elevated (390 U/L)	Balanced
8. M	4.5 y/32 y	Followed	22	122/61	N/A	121	178	N/A	None; CK normal	Balanced
9. M	20 m/31 y	followed	22.7	38/21	N/A	118	141	5.6	None	Balanced
10. F	8 y/33 y	Followed	24	34/21	N/A	40	130	N/A	None	Balanced
11. M	2 y/20 y	Followed	21.2	61/33	12	487	303	6.6	DM type I diagnosed at 4 y	Moderate compliance/DM treated with insulin
12. M	2 y/21 y	Followed	16	32/22	N/A	86	121	2.4	None	Balanced
13. M	4 m/18 y	Followed	N/A	N/A	N/A	N/A	N/A	N/A	N/A	Follow-up: 6 years
14. M	3.5 y/22 y	Followed	N/A	N/A	N/A	N/A	N/A	N/A	N/A	Follow-up: 4 years
15. M	N/A/18 y	Followed	N/A	N/A	N/A	N/A	N/A	N/A	N/A	Follow-up: 8 years
**GSD type IX**										
16. F	3 y/33 y	Followed	25.4	48/32	N/A	91	201	N/A	HL/ Episodes of hypoglycemia in childhood	Balanced/ When coming of age—normal diet

Abbreviations: BMI—body mass index; LA—lactate acid; TG—triglycerides; TC—total cholesterol; UA—uric acid; CK—creatine kinase; CS—cornstarch; CH—carbohydrates; N/A—not available; HL—hyperlipidemia. Reference values: ALT < 45 U/L; AST < 35 U/L; LA 4.5–19.8 mg/dL; TG < 90 mg/dL; TC < 170 mg/dL; UA 3.4–7.0 mg/dL; CK < 195 U/L (men), < 170 U/L (women). Legend: high-carbohydrates and low-fat diet: 60–70% calories from carbohydrates, 10–15% calories from protein, and the remaining calories from fats; high-protein diet: 2–3 g of protein/kg; GSD III/VI/IX diet: low-fat, high-protein diet; moderate metabolic homeostasis—mild biochemical abnormalities, not severe organic complications.

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
