# Peer review of "Over 20-Year Follow-up of Patients with Hepatic Glycogen Storage Diseases: Single-Center Experience"

_diagnostics, 2020, doi:10.3390/diagnostics10050297_

Round 1
Reviewer 1 Report
Even though the authors ahve taken into account many suggestions, remarks, and criticisms, this paper still appears to be confuse, and still contains many mistakes and several imprecisions.
First, the introduction, which has been changed, is now far too long for such a paper. Many mistakes are still present within this paragraph:
line 44: thrombocytopenia is not common in GSD I patients
line 53: in GSD type I, ketogenesis has not the time to be triggered, the fasting tolerance being too short
lines 55 to 58: the text is not clear, and hard to follow between liver, kidney, etc..
line 77: GSD type III patients often need cornstach when they are children
line 90: prolonged fasting tests are not usually performed and are not useful (and even dangerous) for differentiating between type I and type III
line 95: enzyme assays are performed in blood cells for diagnosisng type III, Vi and IX, but as glucose-6-phosphatase is not expressed in blood cells, it is not used for type I (the same mistake is repeated line 138)
In the diagnosis paragraph, it is hard to follow the text which is not clearly written (one paragraph lines 159 to 164 talking about complications, then one dealing with treatments, and then back to complications from line 173 to line 179
In the discussion, line 212, it is not completely true to write that in GSD III adults , liver involvement is asymptomatic (up to 12 to 15% of adults develop fibrosis , cirrhosis).
Line 246, alphafoetoprotein is not a reliable marker for detecting malignant transformation.
Author Response
We thank the Reviewers for reading the manuscript carefully, spotting all the unclear passages and pointing them out. We improved the manuscript, incorporating answers to Your questions.
First, the introduction, which has been changed, is now far too long for such a paper. Many mistakes are still present within this paragraph:
line 44: thrombocytopenia is not common in GSD I patients
It has been deleted.
line 53: in GSD type I, ketogenesis has not the time to be triggered, the fasting tolerance being too short
It was corrected as advised.
lines 55 to 58: the text is not clear, and hard to follow between liver, kidney, etc..
It was corrected as advised.
line 77: GSD type III patients often need cornstach when they are children
It was corrected as advised.
line 90: prolonged fasting tests are not usually performed and are not useful (and even dangerous) for differentiating between type I and type III
This information was added to the manuscript.
line 95: enzyme assays are performed in blood cells for diagnosisng type III, Vi and IX, but as glucose-6-phosphatase is not expressed in blood cells, it is not used for type I (the same mistake is repeated line 138)
It was corrected as advised.
In the diagnosis paragraph, it is hard to follow the text which is not clearly written (one paragraph lines 159 to 164 talking about complications, then one dealing with treatments, and then back to complications from line 173 to line 179
The section about diagnosis has been more clearly written.
In the discussion, line 212, it is not completely true to write that in GSD III adults , liver involvement is asymptomatic (up to 12 to 15% of adults develop fibrosis , cirrhosis).
It was corrected as advised.
Line 246, alphafoetoprotein is not a reliable marker for detecting malignant transformation.
The information about serum AFP has been deleted.
Reviewer 2 Report
Thank-you to the authors for providing more information regarding methods of evaluation of patients.
The more significant issues of the manuscript were not addressed in the revised manuscript. What does the manuscript add to the literature? How does it differ from that already published, compelling a reader to add new knowledge to their repertoire of GSD/metabolics knowledge.
I continue to have concerns regarding table 1 as not being de-identified enough
Author Response
The more significant issues of the manuscript were not addressed in the revised manuscript. What does the manuscript add to the literature? How does it differ from that already published, compelling a reader to add new knowledge to their repertoire of GSD/metabolics knowledge.
We thank the Reviewer for his/her opinion.
Our study presented a long-term (over 20 years) follow-up of GSD patients provided by one referral center. Such long-term follow-up is sparse in the GSD literature.
We provided very useful information regarding patients’ compliance and the disease course. All of patients were closely and regularly followed-up, even as adults. Due to that, patients’ adherence improved (they are taught how to manage the diet and control themselves), and by transition (at the age of 18 years when pediatric healthcare can no longer be provided, and internal/adult medicine care is delivered) the vast majority of them had balanced metabolic homeostasis.
I continue to have concerns regarding table 1 as not being de-identified enough.
We do not agree with this opinion. The Table is clearly written for the reader and the clue information about the diagnosis and disease course are provided.
Round 2
Reviewer 1 Report
The authors have carbon-copied my remarks in their revised version of the paper. This is not enough to make this paper suitable for publication
This manuscript is a resubmission of an earlier submission. The following is a list of the peer review reports and author responses from that submission.
Round 1
Reviewer 1 Report
Dr Szymanska and colleagues are reporting their experience with 30 adult patients suffering from different types of hepatic GSDs. This is an interesting and important subject.
However, this paper , in its form, brings very little new informations. Several criticisms may be made.
First, the paper is not clearly written, and sometimes very confused. In particular, the introduction section should be totally changed, in order to clarify between clinical, biological and genetic data.
Second, many mistakes are written such as , line 58 (introduction), "night glucose infusions are no longer provided", whereas nocturnal gastric drip feeding are still used and useful infants and young children. Another important mistake is read line 100 "enzymatic testing from blood leukocytes" for the diagnosis of GSD Ia (Glucose-6-phosphatase is not expressed in red blood cells).
Third, the authors should write how the diagnosis of GSD has been made, including the results of genetic studies.
Fourth, how were detected HCA in patients with GSD I (ultrasound, MRI?)?
Fifth : it would be important to know if some patients have been given médications (allopurinol, fibrates, others?....)
Sixth: Nothing is written regarding peripheral myopathy in GSD III patients, while few patients have increased CPK levels . It would also be interesting to know more détails for patients with normal CPK levels
Reviewer 2 Report
Although this study is interesting, it did not belong to the scope of this journal. More specific journal may be the better choice.
Reviewer 3 Report
While I would like to commend the authors for reviewing a cohort of very rare disorders the paper requires major revisions prior to any publication.
Importantly, what does this paper add to the literature and breadth of knowledge about this patient population? Is there an aspect of care that was different, the completeness of the documentation, does it have a sub population that are part of a "founder effect" and their course differs from that found in the accepted conventional wisdom about the onset, course and management of these patients. The novel aspect of their work needs to be highlighted in the introduction to demonstrate why this review was completed.
The paper also requires a review of sentence structure, grammar and punctuation.
It would be very helpful to understand this patient population better, where are the referrals from, how does this compare to other European Metabolics referral centres. How many patients were referred or diagnosed over this time period and how many were lost to follow-up and why or hypothesis why.
The table has too much identifying information, as it provides individual ages of patients and then their medical history. For a single referring site and rare disorders this could be identify for individual patients. A consolidated table based on GSD type would be considered more appropriate, ie range of years of follow up, with mean, range of age at diagnosis with a mean value.
The discussion could be strengthened with more of a literature review, again providing an opportunity to highlight why this paper is novel. Perhaps a table outlining your findings compared to previously published work. All would strengthen the paper and its addition to the body of literature on GSD
Were there any liver biopsy results to add to the information, or serial fibroscans, or other markers other than liver transaminases?
Large single centre experience is important but in needs to be framed on how it is novel, no provide too much individual patient information and have a greater literature comparison. With major revisions this information could add to the body of knowledge in this area.